# An Efficient Propagation System through Root Cuttings of an Ecological and Economic Value Plant—*Broussonetia papyrifera* (L.) L’Hér. ex Vent

**DOI:** 10.3390/plants11111423

**Published:** 2022-05-27

**Authors:** Jintuo Zou, Jiana Lin, Bingnan Zhang, Qingmin Que, Junjie Zhang, Youli Li, Yonggui Liu, Xiangbin Zhou, Xiaoyang Chen, Wei Zhou

**Affiliations:** 1Guangdong Key Laboratory for Innovative Development and Utilization of Forest Plant Germplasm, South China Agricultural University, Guangzhou 510642, China; jintuozou@163.com (J.Z.); jianalin@stu.scau.edu.cn (J.L.); zbn@stu.scau.edu.cn (B.Z.); qmque@scau.edu.cn (Q.Q.); zhangjunjie3168@163.com (J.Z.); 2114965524@stu.scau.edu.cn (Y.L.); wsr199843@163.com (Y.L.); 2State Key Laboratory for Conservation and Utilization of Subtropical Agro-Bioresources, Guangzhou 510642, China; 3College of Forestry and Landscape Architecture, South China Agricultural University, Guangzhou 510642, China; 4Guangdong Engineering Technology Research Center of Agricultural and Forestry Biomass, South China Agricultural University, Guangzhou 510642, China; 5Lianshan Forest Farm of Guangdong (Administration of Guangdong Yingyangguan National Forest Park), Qingyuan 513200, China; zxiangbin@foxmail.com

**Keywords:** *Broussonetia papyrifera*, root, vegetative propagation, Hoagland nutrient solution

## Abstract

*Broussonetia papyrifera* (L.) L’Hér. ex Vent. has considerable economic and ecological value and a long history of use in China. In this paper, root cuttings were used as the material to establish an efficient vegetative propagation of *B. papyrifera*. The results revealed that root segments with a diameter of 1.5~2.0 cm and a length of 20~30 cm were most suitable for shoot regeneration, as these segments had the highest adventitious shoot induction rates (93.3%), strongest adventitious shoots, and highest multiplication coefficients (7.07). With regard to the methods used for root burial, a horizontal burial at a depth of 1~3 cm yielded the best results, in this case, the adventitious shoot induction rate can reach 86.7%. The best substrate combination was perlite: peat: coconut chaff = 1:1:1 (*v*/*v*/*v*), wherein the adventitious shoot induction rate can reach 75.6%. The best sterilization method was mixing soil with carbendazim and soaking the root sections in carbendazim for 30 min, wherein the adventitious shoot induction rate can reach 77.8%. Adding 0.2 mg/L naphthaleneacetic acid (NAA) to 1/4 Hoagland nutrient solution significantly improved the rooting rate of adventitious shoots to 82.2%, and the survival rate of the acclimatized plants was more than 90.0%.

## 1. Introduction

The vegetative reproduction of plants refers to the cultivation of new plants by using the regenerative abilities of roots [1], leaves [2], and stems [3], and it is achieved by methods such as cutting [4,5], grafting [6,7], layering [8,9], and tissue culturing [10], among others. Compared with reproduction by seed, vegetative propagation can shorten the breeding period, making it suitable for large-scale propagation and the maintenance of ideal genetic characteristics [11]. Therefore, it is the preferred propagation method for many species [12,13]. For example, vegetative propagation of Trillium govanianum Wall ex D. Don is achieved by positioning rhizomes horizontally in plastic bags containing 50% organic compost, 25% soil, and 25% sand [14], whereas adventitious stem cuttings of Cycas achieve higher rooting rates with sealer [15].

*Broussonetia papyrifera* (paper mulberry) is a highly valued plant that is native to Asia but that is now distributed widely throughout the world [16,17]. The leaves, bark, roots, fruit, and branches of *B. papyrifera* can be used in aquaculture, industry, and medicine. For example, feeding *B. papyrifera* silage to beef cattle improves their final body weight, dry matter intake, and feed conversion ratio [18], and improves the antioxidant capacity and immunity of Holstein dairy cows [19]. Using ionic liquid pretreatment and coprecipitation magnetization technology, leaves of *B. papyrifera* were successfully transformed into a new type of magnetic adsorbent [20]. Due to its strong root system, the species is considered suitable for soil and water conservation. *B. papyrifera* is also highly stress resistant [21] and can grow in soils polluted with heavy metals, and therefore may play an important role in ecological restoration [22,23]. In short, *B. papyrifera* has great practical value.

Although *B. papyrifera* can be propagated by seeds, there are some disadvantages in seed propagation, such as the inability to maintain the genetic characteristics. Root cutting is a kind of vegetative propagation which uses the totipotency of plant cells and use plant roots as materials. It entails burying the plant roots in the substrate, and then separating the adventitious shoots for rooting culture after the adventitious shoots grow to a suitable height. The root biomass of *B. papyrifera* is large, so the root is an excellent propagation material. Plant tissue culture of *B. papyrifera* has been reported [24], but it requires operation under sterile conditions with advanced technical requirements [25]. However, propagation through root cuttings does not require a sterile environment so it is a convenient method of operation. Previously, we collected cuttings and root segments of *B. papyrifera* and then stored them under the same conditions, and we found that root cutting is more suitable to the long-distance preservation of the resources of *B. papyrifera*. The cost of root cutting is low, and the multiplication coefficient is high, but there is no efficient process through root cuttings of *B. papyrifera.* Therefore, this paper studies the effects of different treatments on root propagation and explores the optimal conditions for root propagation. At the same time, our team also designed multiple nurseries to improve the whole root propagation system and greatly improve the propagation rate of *B. papyrifera*.

## 2. Materials and Methods

### 2.1. Materials

The experiment was conducted at South China Agricultural University between June and October 2020. The local climate is a subtropical monsoon climate. Plant materials were collected from the Ningxi Experimental Base (113.64° E, 23.24° N), and the provenance was Guangzhou. The material used was a mixture of root segments of several superior trees. Robust roots, 0.5~2.0 cm in diameter and 5~30 cm long, were selected. After harvesting, roots were immediately wrapped in a wet towel and placed in a polystyrene box filled with ice water to prevent moisture loss. Roots were propagated in seedling trays (50 cm long × 25 cm wide × 10 cm deep), in a matrix comprising peat, coconut bran, perlite, and vermiculite. The experiment was carried out in a greenhouse, the temperature was controlled at 26 ± 2 °C, and the light source was natural light. The substrate of the seedling tray was thoroughly watered before root embedding, and then each seedling tray was watered with 500 mL water every day. All plant growth regulators are from Phyto Technology Laboratories (Lenexa, KS, USA). In addition, the experiments of each treatment contained 15 root segments, and all experiments were repeated three times.

### 2.2. Experimental Design

#### 2.2.1. Effects of Segment Size

A two-factor experimental design was used to test the effects of root segment length and diameter on propagation. A total of 16 treatments were designed, which were A (length = 5 cm, diameter = 0.5~0.8 cm), B (length = 10 cm, diameter = 0.5~0.8 cm), C (length = 20 cm, diameter = 0.5~0.8 cm), D (length = 30 cm, diameter = 0.5~0.8 cm), E (length = 5 cm, diameter = 0.8~1.1 cm), F (length = 10 cm, diameter = 0.8~1.1 cm), G (length = 20 cm, diameter = 0.8~1.1 cm), H (length = 30 cm, diameter = 0.8~1.1 cm), I (length = 5 cm, diameter = 1.1~1.5 cm), J (length = 10 cm, diameter = 1.1~1.5 cm), K (length = 20 cm, diameter = 1.1~1.5 cm), L (length = 30 cm, diameter = 1.1~1.5 cm), M (length = 5 cm, diameter = 1.5~2.0 cm), N (length = 10 cm, diameter = 1.5~2.0 cm), O (length = 20 cm, diameter = 1.5~2.0 cm), P (length = 30 cm, diameter = 1.5~2.0 cm). Following disinfection (soaked in 0.1% carbendazim for 30 min), the root segments were placed in the seedling tray with mixed substrate (perlite:peat soil = 1:1 (volume ratio)) by horizontal burying and covering with 3cm of soil. After 15 days, the adventitious shoot induction of segments was assessed, and after 30 days, average shoot height and absorbing root formation rates were measured.

#### 2.2.2. Effects of Burial Method and Depth

Segments 0.8~1.1 cm in diameter and 20 cm in length were used in this experiment. Since there are more roots with a diameter of 0.8~1.1 cm in the root system of *B. papyrifera*, combined with the adventitious shoot induction rate and multiplication coefficient obtained from the pre-experiment, we selected the root segment with diameter of 0.8~1.1 cm and length of 20 cm as the material. Horizontal burial entailed placing root segments horizontally at 2 cm intervals in seedling trays and covering segments with 3 cm of soil. In the vertical burial treatment, the base of the root segment was inserted vertically into a pot with a diameter of 24 cm and a depth of 30 cm and then covered with 1 cm of soil. For the experiment involving root-covering thickness, four thicknesses (0 cm, 1 cm, 3 cm, and 5 cm) were set.

#### 2.2.3. Effects of Different Substrates

The effects of substrate type on adventitious shoot induction rates and multiplication coefficients were studied via horizontal burial method. There are four types of substrates: Perlite:peat soil:coconut chaff (*v*/*v*/*v* = 1:1:1), Perlite:peat soil:vermiculite (*v*/*v*/*v* = 1:1:1), Vermiculite:peat soil:coconut chaff (*v*/*v*/*v* = 1:1:1), and Perlite:vermiculite:coconut chaff (*v*/*v*/*v* = 1:1:1).

#### 2.2.4. Effects of Disinfection Method

Different disinfection methods were used to treat the substrate and root segments, and six treatment combinations were established to identify the best disinfection method (Table 1). In this experiment, the root segment with diameter of 0.8~1.1 cm and length of 20 cm was placed in the seedling tray by horizontal burying and covering with 3 cm of soil. Substrate treatments were prepared as follows: substrate was first moistened with water, then sprayed with either 0.1% potassium permanganate solution or 0.1% carbendazim suspension on the surface (sprayed with 500 mL per seedling tray), mixed thoroughly, covered with black plastic film, and sealed. After one week, substrates were used in root propagation. In the root segment treatment, root segments were rinsed 2–3 times with clean water and immersed for 30 min in either a 0.1% carbendazim suspension, a 0.1% chlorothalonil suspension, or distilled water.

#### 2.2.5. Rooting of Adventitious Shoots

Adventitious shoots can be transplanted together with the root segments, but this will reduce the multiplication coefficient, so the adventitious shoot must be cut to induce its rooting. The adventitious shoots were separated from the root segments when they were 8~10 cm long and had 4~6 true leaves. In addition, the experiment was repeated three times and each treatment contained 15 adventitious shoots. NAA (naphthaleneacetic acid) (0, 0.1, 0.2, and 0.5 g/L) and IBA (Indole-3-butyric acid) (0, 0.1, 0.2, and 0.5 g/L) were added to the nutrient solution to induce rooting and obtain complete plants. The nutrient solution used in hydroponic culture was 1/4 Hoagland nutrient solution [26]; this was aerated using an air pump to prevent rot.

#### 2.2.6. Histological Observation of Adventitious Shoot Rooting

During histological observations, because the root segments were thick and lignified, they could not be cut using conventional methods, so we used a hard-tissue-slicing technique that is used in the field of medicine to cut bones [27].

#### 2.2.7. Acclimatization of Rooted Plantlets

The plants were carefully removed from the nutrient solution and the roots rinsed with water to remove the nutrient solution, taking care to avoid damaging roots, and then the plants were transplanted into moist mixed substrate (perlite:peat soil = 1:1). The substrate was poured through with water before transplanting plantlets. The transplanted plantlets were placed into a growth chamber, and the plantlets were kept under cool-white light (1500 µmol/m^2^/s) with a 12 h photoperiod and at a temperature of 26 ± 2 °C. Relative humidity was maintained at 70% during the first week and 50% during the second week. After two weeks in a growth chamber, plantlets were transplanted into flowerpots (15 cm diameter × 20 cm heigh) with a mixed substrate (perlite:peat soil = 1:1) and placed outside for a week to adapt to external conditions, and then transplanted into a field covered with black plastic film in September at an average temperature of 26 ± 2 °C. Mulching with black plastic film prevented weeds from affecting plantlets’ growth. During transplantation, a hole was cut in the plastic film and another hole was dug about 10 cm deep, and then the plantlets were planted with row spacings of 50 × 50 cm. The survival rate of plantlets was counted after one month.

### 2.3. Data Analyses

Data were analyzed using SPSS 19.0 (SPSS Inc., Chicago, IL, USA). Path analysis was carried out with the agricolae package in R version 4.0 [28] to analyze the linear relationship between independent variables and dependent variables. The Duncan multiple range comparison was used to assess differences among three or more groups, whereas *t*-tests were used to assess differences among two groups. Data were visualized using SPSS 19.0, R version 4.0, and Microsoft Excel (Microsoft Corporation, Redmond, WA, USA).
Adventitious shoot induction rate (%) = Number of budding explants/Number of explants × 100.
Net multiplication coefficient (number of adventitious shoots per budding root segment) = Total number of shoots/Number of sprouting segments.
Total multiplication coefficient (number of adventitious shoots per root segment) = Total number of shoots/Total number of root segments.
Absorbing root formation rate (%) = Number of root segments with absorbing roots/Number of sprouting root segments × 100.
Rooting rate (%) = Number of adventitious shoots rooting/Number adventitious shoots × 100.

## 3. Results

### 3.1. Effects of Segment Size on Adventitious Shoot Induction Rates and Multiplication Coefficients

Both the length and diameter of the root segments had significant effects on shoot production and the total multiplication coefficients (Table 2). The lowest adventitious shoot induction rate (20.0%) was observed in Test A (length = 5 cm, diameter = 0.5~0.8 cm), whereas the highest rate (93.3%) was observed in Test O (length = 20 cm, diameter = 1.5~2.0 cm). The net multiplication coefficient was lowest in Test E (length = 5 cm, diameter = 0.8~1.1 cm), at 1.50, and highest in Test P (length = 30 cm, diameter = 1.5~2.0 cm), at 8.13. The lowest total multiplication coefficient (0.33) was observed in Test A (length = 5 cm, diameter = 0.5~0.8 cm), and the highest (7.07) in Treatment Test P (length = 30 cm, diameter = 1.5~2.0 cm).

At a constant diameter, a longer segment length was associated with higher induction rates. Adventitious shoot induction rates were 20.0~53.3% for 5-cm-long segments, 26.7~80.0% for 10-cm-long segments, 46.7~93.3% for 20-cm-long segments, and 66.7~86.7% for 30-cm-long segments. In addition, the induction rates also increased with the segment diameter. Results (Table 2) indicate that at a constant root diameter, longer segments had higher multiplication coefficients; similarly, when the root length was fixed, larger diameters were associated with higher multiplication coefficients.

Two-way analysis of variance (ANOVA) revealed that the length and diameter of the root segments significantly affected shoot induction rates as well as the net and total multiplication coefficients (*p* < 0.01). As evident in Table 3, the length and diameter of the roots had the same contribution to the induction rate of the adventitious shoots, while the effect of root length on the multiplication coefficient was greater than that of root diameter.

However, we also observed other trends. When the segment diameter was 0.8~1.1 cm and 1.5~2.0 cm, the adventitious shoot induction rate of the segment with a length of 20 cm was higher than that of the segment with a length of 30 cm. To explore this phenomenon, we conducted a further analysis. At segment lengths of approximately 20 cm, total multiplication coefficients appeared to converge for the segments with diameters of 0.8~1.1 cm and 1.1~1.5 cm (Figure 1a), whereas the net multiplication coefficient of the former was much higher than that of the latter (Figure 1b). For 20-cm-long and 30-cm-long segments, the net multiplication coefficient of segments with a diameter of 0.8~1.1 cm was higher than that of the segments with a diameter of 1.1~1.5 cm. For segments with diameters between 0.8 and 1.5 cm, the induction rate of 20-cm-long segments was higher than that of the 30-cm-long segments (Figure 1c).

### 3.2. Effects of Segment Size on Absorbing Roots Formation Rate and the Growth of Adventitious Shoot

At a constant diameter, longer root segments had a higher absorbing root formation rate (Table 4). The absorbing root formation rate was relatively high for 30-cm-long segments, at 70.0~88.90%. Conversely, at a constant segment length, larger diameter segments were associated with lower rooting rates of the absorbing root. The rooting rate (66.7~88.90%) of absorbing roots was significantly higher for segments with a diameter of 0.5 cm than the other three diameters. The average height of the adventitious shoots was 10.50~14.80 cm for 5-cm-long segments, 12.47~21.16 cm for 10-cm-long segments, 16.71~22.81 cm for 20-cm-long segments, and 17.94~25.98 cm for 30-cm-long segments. As for the absorbing root formation rate and the growth of adventitious shoots, the analysis of variance showed that the root segment length, diameter, and interaction between length and diameter of the *B. papyrifera* had extremely significant effects on the monthly average height of plantlets (*p* < 0.01), while the root segment length and diameter had significant effects on the absorbing root’s formation rate (*p* < 0.05).

The results suggest (Table 4) that longer segments are more conducive to the production of absorbing roots and the growth of adventitious shoots. Larger diameter segments were more conducive to shoot growth but had a lower absorbing root formation rate. 

### 3.3. Effects of Root Burial Method and Burial Depth

The results indicate that the best burial depth is 1~3 cm (Table 5). Due to the high summer temperatures during the experimental period, exposed root segments experienced severe water loss and desiccation, destroying their capacity. The adventitious shoot induction rates are 73.3% for burial at 1 cm depth, 86.7% for burial at 3 cm depth, and 43.1% for burial at 5 cm depth.

The results showed that horizontal burial was superior to vertical burial with regard to both adventitious shoot induction rates and total multiplication coefficients. From Table 6, we can see that different root burying methods had significant effects on the multiplication coefficient and the adventitious shoot induction rate. Each root segment contains numerous adventitious growth points, each of which can differentiate into adventitious shoots. In contrast, when segments are positioned vertically, white callus is only produced at the top of the root segment, resulting in few growth points and a lower multiplication coefficient. Horizontal burial is also more convenient, more cost-effective in that it requires less soil, and enables an easier excavation of adventitious shoots, making it the best overall burial method.

### 3.4. Effects of Substrate

The results (Table 7) of the comparison of the four substrate types indicate that combination A (Perlite:peat soil:coconut chaff = 1:1:1) yields the highest adventitious shoot induction rate (75.6%) and its absorbing root formation rate was 70.7%. Combination D (Perlite:vermiculite:coconut chaff = 1:1:1) had the highest absorbing root formation rate, but its adventitious shoot induction rate was the lowest, at only 31.1%.

### 3.5. Effects of Disinfection Methods on Adventitious Shoot Induction Rates

The best disinfection method (Table 8) involved combining the soil with 0.1% carbendazim solution and soaking the root sections with 0.1% carbendazim solution for 30 min. This method effectively inhibited the growth of fungi and microorganisms in soil. Carbendazim and chlorothalonil had highly significant disinfection effects. While potassium permanganate and carbendazim both had effects on soil, the effects of potassium permanganate were less pronounced, potentially due to leaching after watering.

### 3.6. Effects of Different Plant Growth Regulator Concentrations on Rooting of Adventitious Shoots and Acclimatization

Adding different types and concentrations of plant growth regulators into a hydroponic nutrient solution had significant effects on the rooting of adventitious shoots, and the concentrations set in this experiment produced adventitious shoot rooting. Within a certain concentration range (0–0.5 g/L), adding different concentrations of naphthaleneacetic acid (NAA) and indole-3-butyric acid (IBA) improved the rooting rates of adventitious shoots. Adding 0.2 g/L NAA yielded the highest rooting rate and resulted in a strong main root with numerous lateral roots. Rooting rates were also enhanced by the addition of 0.1 g/L NAA and 0.5 g/L IBA, but the root system was less developed than with 0.2 g/L NAA. Consequently, the best plant growth regulator concentration to promote rooting is 0.2 g/L NAA (Figure 2). After transplanting to the field, if the plantlets grow new terminal buds, it indicates the survival of the transplant. One month later, propagated plants had a high survivorship rate (>90%) and healthy growth.

Ongoing observations indicated that the root propagation of *B. papyrifera* can be divided into five stages: adventitious shoot formation (Figure 3a), shoot growth (Figure 3b), the growth of new absorbing roots on the root segments (Figure 3c), the growth of new roots on the plant, and acclimatization and transplantation (Figure 3d).

### 3.7. Histological Observation of Adventitious Shoot Rooting

By observing the sprouting process and the production of new absorbing roots using paraffin sections, we found that the budding of adventitious roots differed from that of adventitious shoots in two ways. First, when adventitious shoots are formed, the expanded surface of the bud is visible to the naked eye (Figure 4a), and the epidermal cells in the bud are became significantly larger (Figure 4b). Meanwhile, the surface meristem begins to form bud primordia through anticlinal and pericyclic division (Figure 4c). Second, through the observation of the paraffin sections, we found that the adventitious root primordium began in the vascular cambium, the meristematic cells of which then divided into a growth cone (Figure 4d).

## 4. Discussion

The overall results showed that the size of the root segment, the method and depth ofroot burial, the type of substrate, the method of sterilization, and plant growth regulators had influence on the propagation of *B. papyrifera* root segment. In the following, we further analyze the data based on the experimental results and discuss our findings and our preliminary research hypotheses.

Previous articles explored the influence of root length and root diameter on the budding rates for the root propagation of *Broussonetia kazinoki* SIER. [29]. These articles concluded that the longer the root segment, the larger the root diameter and the higher the budding rate. Since we performed the interaction experiment of root length and root diameter, we thus used path analysis and other data analysis methods to draw some different conclusions. We found that the number of shoots was directly related to the number of adventitious shoot primordia in each segment. Adventitious shoot primordia are unevenly distributed in the root cambium and have an active meristematic capacity. However, the results showed that the adventitious shoot induction rate (93.3%) and multiplication coefficient (6.8) were the highest in the root segments of *B. papyrifera* with a diameter of 1.5~2.0 cm and a length of 20~30 cm. In production, the root segment with a diameter of 1.1~1.5 cm (20~30 cm) is more suitable because the amount of root segment material in this specification is larger and easier to obtain than the root segment with a diameter of 1.5~2.0 cm, and the adventitious shoot induction rate and multiplication coefficient can reach 88.7% and 5.53, respectively. Therefore, longer segments typically have more adventitious shoot primordia, resulting in higher induction rates and multiplication coefficients. At the same time, it was also reported that adventitious shoots were induced by tissue culture with the root segment as the explant. The induction rate can reach 88.9%, and the total multiplication coefficient can reach 5.02 [24]. Compared with the root cuttings in this experiment, tissue culture can be expanded with less root material. When there is less material, the effect of asexual propagation with tissue culture is better, and the excellent variety is well preserved. However, the technical requirements of tissue culture are higher than those of root cutting, as an aseptic operation is required throughout the entire process. In addition, it needs to be combined with various plant growth regulators to induce regeneration. The root cutting operation is easier; there is no need to maintain a sterile environment, and the plantlet training time is shorter. Therefore, it has better operability in production and is more suitable for application in production.

The characteristics of adventitious shoots are related to the rate of production of the absorbing roots. Adventitious shoots growing on mother roots with new absorbing roots had green leaves and strong stems, in contrast, shoots growing on root segments without new absorbing roots had yellow leaves and thin, weak stems. Therefore, adventitious shoots of thicker roots should be transplanted promptly to prevent the mother roots from inhibiting the absorption of water and nutrients due to the inability to produce new absorbing roots, which may result in withered, yellow, and weak shoots and an increased shoot mortality.

The method of material placement affects the root propagation efficiency of *B. papyrifera*. This phenomenon has also been reported in other plants, such as *Jatropha curcas*, where the regeneration efficiency of horizontal placement is higher than that of vertical placement [30]. This may be related to the polar transport of auxin [31].

It is evident from Table 7 that the induction rate is low when the root section is not covered with soil. In this case, the root segment easily loses water, leading to inactivation. In addition, when segments were covered with 5 cm of soil, root induction rates decreased. Excessively deep burial inhibited respiration, and even when bud primordia have formed, new shoots had difficulty breaking through the soil. In contrast, covering segments with 1~3 cm of soil created ideal light, temperature, and ventilation conditions, this not only benefited the root growth, but also reduced the obstruction to the emergence of new shoots, thus increasing the root induction rate and total multiplication coefficient.

The results of the substrate test showed that the combination of perlite, peat soil, and coconut chaff (1:1:1) had a better effect. Perlite has a strong water absorption capacity, good air permeability, a honeycomb structure, and porous adsorption, which can influence soil fertility [32]. Coconut chaff has a better air-water ratio, volumetric weight, and a suitable pH, making it conducive to water and air exchange in the substrate. Peat soil [33] is rich in humus, which can provide nutrients for plant growth, and has a strong water and fertilizer retention capacity, which can reduce water loss. A combination of the three substrates may significantly improve the air permeability, water absorption, water holding capacity, and fertilizer efficiency of substrate. The best proportion of the three can be further refined in future work.

By observing the histology of the paraffin section of adventitious shoot germination in the root segment of *B. papyrifera*, we found that the epidermal cells in the bud were substantially enlarged, and the cytoplasm was thick, which resulted in darkened coloring after staining with safranin fast green. Moreover, epidermal cell division is gradually activated, proliferation accelerates, and developed into callus with meristem nodules. The surface callus slowly splits to form a new growth center and begins to form bud primordium, and the formation of the bud primordium was similar to that of Lisianthus (*Eustoma grandiflorum* (Raf.) Shinners) [34]. Moreover, we found that the budding of adventitious shoots is exogenous in root segments of *B. papyrifera*, the same as the results reported by Sunanda [35] in *Limnophila indica* (L.) Druce, Subotić [36] in *Centaurium erythreae* Gillib and Shankhamala [37] in *Limonium* hybrid ‘Misty Blue’.

We have carried out the pre-experiment of the adventitious shoot rooting of *B. papyrifera* and found that adventitious shoots could not take root on a solid substrate. To solve this problem, we referred to the hydroponic methods of some plants, such as tomatoes [38], kiwi fruits [39], cucumbers [40] and so on, and decided to use a hydroponic method to induce adventitious buds to take root. The results revealed that 1/4 Hoagland nutrient solution was the best culture medium, resulting in appropriate concentrations of inorganic salts and nutrients, high water and nutrient transport efficiency, and high shoot survivorship prior to rooting. Previous studies have shown that IBA can increase root initiation in cuttings of *Eucalyptus nitens* [41], induce adventitious root formation in apple plants [42], and NAA can promote the rooting of *Moringa oleifera* Lam. [2]. NAA and IBA were used to induce adventitious root production. This study concluded that the rooting rate of adventitious shoot could be significantly improved by adding 0.2 g/L NAA to the nutrient solution. During the acclimatization period, there was no need to wipe the buds or apply fertilizer. Transplant survival was more than 90% following acclimatization.

To establish a set of efficient *B. papyrifera* propagation systems, based on this experiment, we can establish a multi nursery supporting plantlet system, which includes a root picking nursery, a root burying nursery, a root accelerating nursery, and an adult plantlet nursery. The multi nursery supporting plantlet system standardizes the propagation steps of *B. papyrifera* with root segments as materials, and has the advantages of a clear process, a high feasibility, and economic applicability. This system is of great significance for the rapid propagation and large-scale promotion of good varieties of *B. papyrifera* and promoting the industrialization of *B. papyrifera*.

## 5. Conclusions

We conclude that the root segments of *B. papyrifera* with a diameter of 1.5~2.0 cm and a length of 20~30 cm had the largest adventitious bud induction rate (93.3%) and value-added coefficient (6.8). The best burial method was to bury segments horizontally at a depth of 1~3 cm. The best substrate combination was perlite:peat:coconut bran = 1:1:1 (*v*/*v*/*v*). In addition, the best sterilization method was carbendazim mixed with soil + soaking segments in carbendazim for 30 min. Finally, adventitious shoot was induced by a hydroponic cutting method. This study established a standardized technical process of *B. papyrifera* root propagation, which involves: (1) segment collection, (2) induction of an adventitious shoot, (3) growth, (4) hydroponic rooting, and (5) transplantation. To accelerate the large-scale popularization of high-quality clones, a nursery of breeding roots should be established to ensure high quality, high yield, and a minimization of the cost of breeding stock. This will facilitate the long-term preservation of high-quality clones, ensure clone purity, and maintain the juvenility of the breeding materials, thus avoiding the negative effects of position and age. 

## Figures and Tables

**Figure 1 plants-11-01423-f001:**
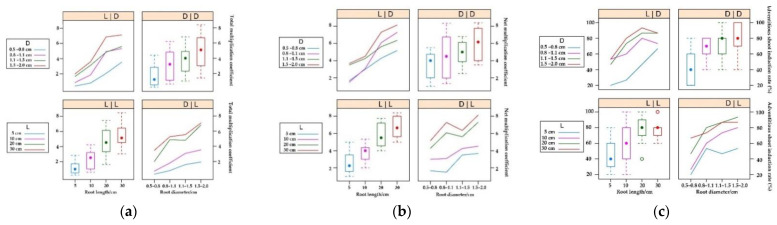
Main effects and 2-way interactions: (**a**) main effects and 2-way interactions of total multiplication coefficient; (**b**) main effects and 2-way interactions of net multiplication coefficient; (**c**) main effects and 2-way interactions of adventitious shoot induction rate. L: root length, D: root diameter, L/L: the main effect of root length, D/D: the main effect of root diameter, L/D and D/L: the interaction between root length and root diameter.

**Figure 2 plants-11-01423-f002:**
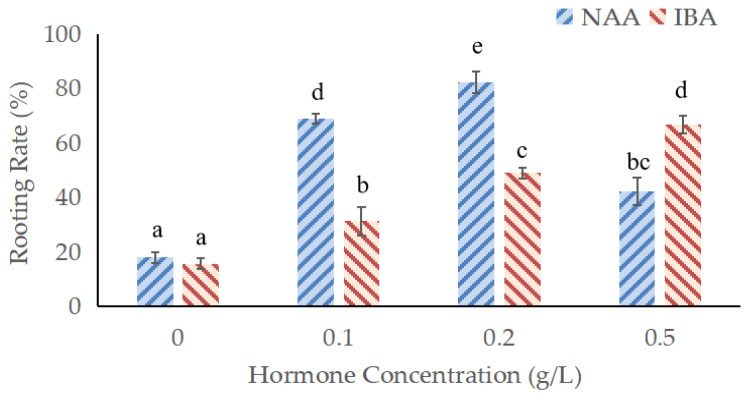
Effects of different plant growth regulator treatments on adventitious shoot rooting of *Broussonetia papyrifera*. Means with the same letters in the same columns are not significantly different from each other at *p* ≤ 0.05 (according to Duncan’s multiple range test).

**Figure 3 plants-11-01423-f003:**
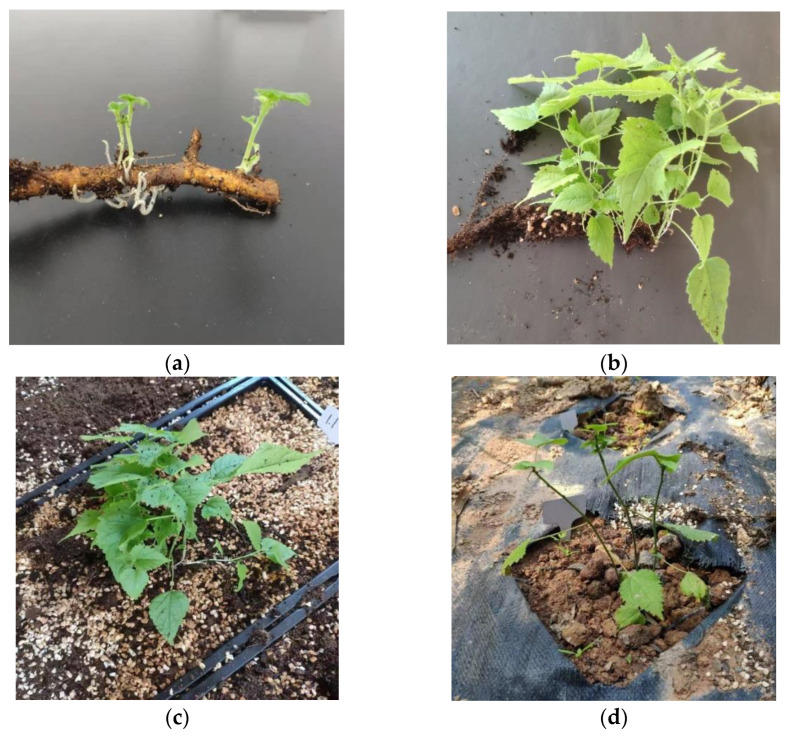
Routine observation on root propagation of *Broussonetia papyrifera*. (**a**) adventitious shoot formation stage of mother root; (**b**) adventitious shoot growth; (**c**) plantlets were transplanted from Hoagland nutrient to mixed substrate (perlite:peat soil = 1:1) for growth; (**d**) growth stage of domestication and the transplanting field.

**Figure 4 plants-11-01423-f004:**
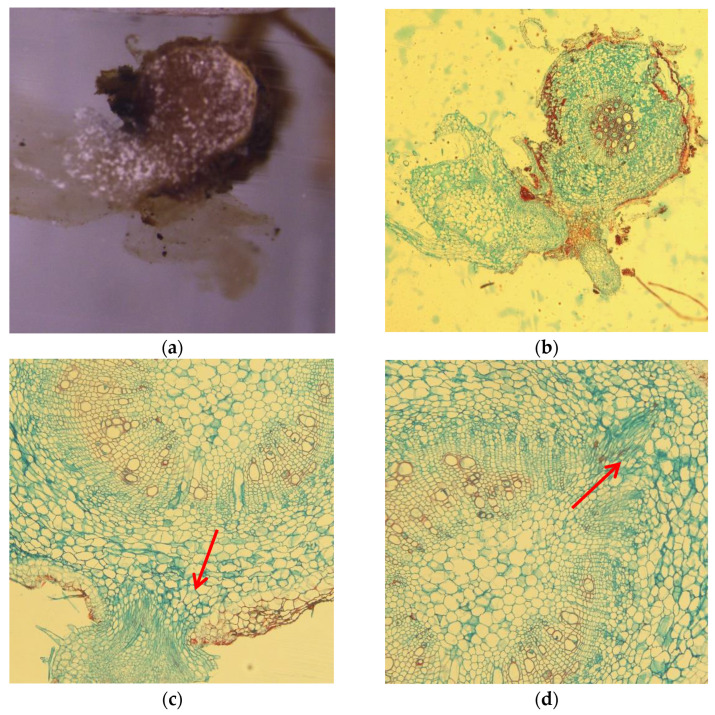
Observation on tissue section of *Broussonetia papyrifera* root propagation. (**a**) the surface of the budding part of adventitious shoot expanded; (**b**) paraffin sections of adventitious buds during budding; (**c**) formation of bud primordium, the red arrow shows the growth direction of the bud primordium; (**d**) budding of adventitious root, the red arrow shows the growth direction of the adventitious root.

**Table 1 plants-11-01423-t001:** Treatment of substrate and root segments with different disinfecting combinations.

Treatment	Treatment of Substrate	Treatment of Root Segments
1	No treatment	Distilled water
2	No treatment	0.1% Carbendazim
3	0.1% carbendazim	0.1% Carbendazim
4	0.1% potassium permanganate	0.1% Carbendazim
5	0.1% carbendazim	0.1% Chlorothalonil
6	0.1% potassium permanganate	0.1% Chlorothalonil

**Table 2 plants-11-01423-t002:** Effects of different root cutting lengths and diameters on root propagation’s adventitious shoot induction rate and the multiplication coefficient of *Broussonetia papyrifera*.

Test	Adventitious Shoot Induction Rate (%)	Net Multiplication Coefficient	Total Multiplication Coefficient	Shoot Characteristics
A	20.0 ± 0.0 ^g^	1.67 ± 0.58 ^g,h^	0.33 ± 0.12 ^f^	thin and weak, single adventitious shoot
B	26.7 ± 11.5 ^f,g^	3.00 ± 1.00 ^f,g^	0.73 ± 0.12 ^e,f^	thin and weak, 2–4 adventitious shoots
C	46.7 ± 11.5 ^e,f^	4.28 ± 0.25 ^d,e,f^	2.00 ± 0.53 ^d,e^	thin and weak, 3–5 adventitious shoots
D	66.7 ± 11.5 ^b,c,d,e^	5.17 ± 0.29 ^c,d,e^	3.47 ± 0.81 ^c^	general, more than 5 adventitious shoots
E	53.3 ± 11.5 ^d,e^	1.50 ± 0.17 ^h^	0.80 ± 0.20 ^e,f^	thin and weak, single adventitious shoots
F	60.0 ± 0.0 ^c,d,e^	3.08 ± 0.88 ^f,g^	1.80 ± 0.60 ^d,e^	thin and weak, 2–4 adventitious shoots
G	80.0 ± 0.0 ^a,b,c^	6.08 ± 1.46 ^b,c^	4.87 ± 1.17 ^b^	general, more than 5 adventitious shoots
H	73.3 ± 11.5 ^a,b,c,d^	7.28 ± 1.35 ^a,b^	5.27 ± 0.83 ^b^	robust, more than 5 adventitious shoots
I	46.7 ± 11.5 ^e,f^	3.50 ± 1.32 ^f^	1.60 ± 0.53 ^e,f^	thin and weak, 3–4 adventitious shoots
J	73.3 ± 11.5 ^a,b,c,d^	4.25 ± 0.90 ^d,e,f^	3.07 ± 0.50 ^c,d^	general, 3–5 adventitious shoots
K	86.7 ± 11.5 ^a,b^	5.62 ± 1.08 ^c,d^	4.80 ± 0.53 ^b^	robust, more than 5 adventitious shoots
L	86.7 ± 11.5 ^a,b^	6.35 ± 0.54 ^b,c^	5.53 ± 1.14 ^b^	robust, more than 5 adventitious shoots
M	53.3 ± 23.1 ^d,e^	3.67 ± 0.29 ^e,f^	1.93 ± 0.76 ^d,e^	thin and weak, 2–4 adventitious shoots
N	80.0 ± 20.0 ^a,b,c^	4.51 ± 0.72 ^d,e,f^	3.53 ± 0.58 ^c^	general, 3–5 adventitious shoots
O	93.3 ± 11.5 ^a^	7.30 ± 0.26 ^a,b^	6.80 ± 0.72 ^a^	robust, more than 5 adventitious shoots
P	86.7 ± 11.5 ^a,b^	8.13 ± 0.23 ^a^	7.07 ± 1.15 ^a^	robust, more than 5 adventitious shoots
LSD				
Length	33.988 **	145.390 **	171.320 **	
Diameter	34.974 **	39.911 **	80.759 **	
length/diameter	0.524	0.176	3.796	

Each value represents the mean ± SD of three replicates. Means followed by the same letter in the same column are not significantly different from each other at *p* ≤ 0.05 level, according to Duncan’s multiple range test. ** represented extremely significant difference at *p* < 0.01.

**Table 3 plants-11-01423-t003:** Path analysis of roots’ length and roots’ diameter in root budding.

Item	Trait	Correlation Coefficient	Direct Effect	Total Contribution	Indirect Effect		
					Total	Through L	Through D
Adventitious shoot induction rate	Length	0.56	0.56	0.31	0		0
	Diameter	0.56	0.56	0.31	0	0	
Net multiplication coefficient	Length	0.80	0.80	0.64	0		0
	Diameter	0.42	0.42	0.18	0	0	
Total multiplication coefficient	Length	0.76	0.76	0.58	0		0
	Diameter	0.52	0.52	0.27	0	0	

**Table 4 plants-11-01423-t004:** Effects of different root sizes on absorbing root formation rate and average monthly plantlets height of *Broussonetia papyrifera*.

Test	Absorbing Root Formation Rate (%)	Monthly Average Shoot Height (cm)	Plantlet Traits
A	66.7 ± 57.7 ^a,b^	10.50 ± 0.82 ^b^	Curly leaves, thinner stems, poor growth
B	83.3 ± 28.9 ^a,b^	12.47 ± 1.23 ^c^	Yellowish green leaves, thinner stems, poor growth
C	88.9 ± 19.6 ^b^	16.71 ± 0.42 ^e^	Greener leaves, thin stems, general growth
D	88.9 ± 19.2 ^b^	17.94 ± 0.14 ^f^	Yellowish green leaves, strong stems, general growth
E	50.0 ± 16.7 ^a,b^	8.98 ± 0.55 ^a^	Yellowish green leaves, thin stems, poor growth
F	66.7 ± 0.0 ^a,b^	16.74 ± 0.91 ^e^	Peak green leaves, strong stems, general growth
G	66.7 ± 14.4 ^a,b^	19.72 ± 1.06 ^g,h^	Peak green leaves, stronger stems, good growth
H	72.2 ± 25.5 ^a,b^	21.97 ± 0.44 ^j,k^	Peak green leaves, stronger stems, better growth
I	44.4 ± 9.6 ^a^	11.14 ± 0.12 ^b^	Yellowish green leaves, strong stems, general growth
J	52.8 ± 21.0 ^a,b^	18.59 ± 0.38 ^f,g^	Peak green leaves, stronger stems, better growth
K	61.7 ± 12.6 ^a,b^	19.35 ± 0.52 ^g^	Peak green leaves, stronger stems, better growth
L	70.0 ± 8.7 ^a,b^	20.58 ± 0.51 ^h,i^	Peak green leaves, stronger stems, better growth
M	41.7 ± 14.4 ^a^	14.80 ± 0.15 ^d^	Yellowish green leaves, strong stems, poor growth
N	52.2 ± 13.5 ^a,b^	21.16 ± 0.61 ^i,j^	Yellowish green leaves, stronger stems, general growth
O	65.0 ± 8.7 ^a,b^	22.81 ± 0.38 ^k^	Peak green leaves, stronger stems, better growth
P	76.7 ± 2.9 ^a,b^	25.98 ± 1.29 ^l^	Peak green leaves, stronger stems, better growth
LSD			
Length	3.330 *	237.545 **	
Diameter	3.386 *	195.209 **	
length/diameter	0.113	11.423 **	

Each value represents the mean ± SD of three replicates. Means followed by the same letter in the same column are not significantly different from each other at *p* ≤ 0.05 level, according to Duncan’s multiple range test. * represented significant difference at *p* < 0.05, ** represented extremely significant difference at *p* < 0.01.

**Table 5 plants-11-01423-t005:** Effects of different root burying methods and thickness of covering soil on adventitious shoots induction rate and multiplication coefficient.

Treatment	Method	Budding Root Segments	Sprout Number	Adventitious Shoot Induction Rate (%)	Total Multiplication Coefficient
**Root burial method**	Horizontal burying	12.33 ± 0.58	68.00 ± 3.00	82.2 ± 3.8	4.53 ± 0.20
	Vertical burying	8.00 ± 1.00	34.33 ± 6.10	53.3 ± 6.7	2.29 ± 0.41
**Soil thickness**	0	1.33 ± 0.58 ^a^	3.33 ± 1.53 ^a^	8.9 ± 3.8 ^a^	0.22 ± 0.10 ^a^
	1 cm	11.00 ± 1.00 ^c^	63.00 ± 2.00 ^c^	73.3 ± 6.7 ^c^	4.20 ± 0.13 ^c^
	3 cm	13.00 ± 1.00 ^c^	71.33 ± 0.51 ^d^	86.7 ± 6.7 ^d^	4.76 ± 0.23 ^d^
	5 cm	7.33 ± 1.53 ^b^	21.00 ± 2.00 ^b^	43.1 ± 8.5 ^b^	1.24 ± 0.16 ^b^

Each value represents the mean ± SD of three replicates. Means followed by the same letter in the same column are not significantly different from each other at *p* ≤ 0.05 level, according to Duncan’s multiple range test.

**Table 6 plants-11-01423-t006:** Variance analysis of different root burying methods on adventitious shoot induction rate and multiplication coefficient.

		Df	Sum of Squares	Mean Square	F	Pr (>F)
Adventitious shoot induction rate	Between groups	1	0.125	0.125	42.25	0.003 *
	Within group	4	0.012	0.003		
	Total	5	0.137			
multiplication coefficient	Between groups	1	7.556	7.556	73.388	0.001 *
	Within group	4	0.412	0.103		
	Total	5	7.968			

* Represented significant difference at *p* < 0.05.

**Table 7 plants-11-01423-t007:** Effects of different substrates on adventitious shoot induction rate.

Substrates (*v*/*v*/*v*)	Number of Budding Root Segments	Adventitious Shoot Induction Rate (%)	Absorbing Roots Formation Rate (%)
Perlite:peat soil:coconut chaff	11.33 ± 1.53 ^c^	75.6 ± 10.2 ^c^	70.7 ± 1.8 ^b^
Perlite:peat soil:vermiculite	8.33 ± 1.53 ^b^	57.8 ± 7.7 ^b^	57.5 ± 6.6 ^a^
Vermiculite:peat soil:coconut chaff	7.33 ± 0.58 ^b^	48.9 ± 3.8 ^b^	58.9 ± 3.1 ^a^
Perlite:vermiculite:coconut chaff	4.67 ± 1.15 ^a^	31.1 ± 7.7 ^a^	72.2 ± 4.8 ^b^

Each value represents the mean ± SD of three replicates. Means followed by the same letter in the same column are not significantly different from each other at *p* ≤ 0.05 level, according to Duncan’s multiple range test.

**Table 8 plants-11-01423-t008:** Effects of different disinfection combinations on root propagation induction rate.

Treatment	Treatment of Substrate	Treatment of Root Segments	Number of Budding Root Segments	Adventitious Shoot Induction Rate (%)
1	No treatment	Distilled water	3.67 ± 0.58 ^a^	24.4 ± 3.8 ^a^
2	No treatment	0.1% Carbendazim	7.00 ± 1.00 ^b^	46.7 ± 6.7 ^b^
3	0.1% carbendazim	0.1% Carbendazim	11.67 ± 0.58 ^c^	77.8 ± 3.8 ^c^
4	0.1% potassium permanganate	0.1% Carbendazim	10.00 ± 1.00 ^c^	66.7 ± 6.7 ^c^
5	0.1% carbendazim	0.1% Chlorothalonil	10.67 ± 2.52 ^c^	71.1 ± 16.8 ^c^
6	0.1% potassium permanganate	0.1% Chlorothalonil	9.33 ± 1.53 ^b,c^	62.2 ± 10.2 ^b,c^

Each value represents the mean ± SD of three replicates. Means followed by the same letter in the same column are not significantly different from each other at *p* ≤ 0.05 level, according to Duncan’s multiple range test.

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
