# Peer review of "An Efficient Propagation System through Root Cuttings of an Ecological and Economic Value Plant—*Broussonetia papyrifera* (L.) L’Hér. ex Vent"

_plants, 2022, doi:10.3390/plants11111423_

Round 1
Reviewer 1 Report
The manuscript, “An effcient propagation system through root cuttings of an ecological and economical value plnat—Broussonetia papyrifera (L.) L´Hér. Ex Vent” presents a new propagation system of adventitious shoots through root cuttings. The authors study the efficiency of this propagation system and evaluate different factors like size and diameter of initial root segments, different induction methods, substrate type, sterilization method, hormone concentration for rooting etc.
Overall the manuscript is well presented and structured, the methodology is appropiated, and all the experiments have been carried out properly. The results provide relevant information for the vegetative propation of this species by using root segments as starting material to obtain a high rate of rooted plantlets.
However, there are a few shortcomings that should be addressed prior to publication. Although the manuscript introduces interesting results, some sections should be improved and rewritten. I found misinterpretation of data as well as inconsistences through the text that need to be addressed .
My concerns are indicated beloww :
Abstract:
Line 24: cuttingd should be replaced by cuttings
Lines 25-26: I found misinterpretation of results regarding to data presented in table 2.
The authors make no reference at any time to the 1.5-2 cm segments, and according to the results in Table 2, they are the ones that produce the most adventitious shoots as well as the highest proliferation coefficients. However, they refer to segments 1.1-1.5 as the most suitable, but provided data from segments with a diameter of 1.5-2 cm (93.3 % for root induction rate as well as proliferation coefficients (7.07).
Line 26: In my opininion the term “root propagation” is not appropiate, since you are no propagating roots. Instead you can say for shoot regeneration.
Lines 27-28: Authors use the comparative adjetive higher, stronger…it should be related/compared to another treatments or segment types. Otherwise, the superlative adjetives should be used
Introduction
Line 57: please insert and space after [21]
Materials and Methods Section
Line 91: And the experiments OF each treatment.
Line 107: The description of some parameters is somehow confusing. In the section 2.2.1 (Effects of segment size) they mentioned the assessment of shoot induction, average shoot height and growth of absorptive roots. However, in the results section, they report the percentage of absorbing /absorptive roots.
Evaluation of shoot characteristis/ quality must be included
Subsection 2.2.3
Te whole sentence should be rewriten. The verb is missing
Subsection 2.2.5
Line 139: in my opinion the plural can be used. Adventitious shoots instead of adventitious shoot
Line 141: Adventititous shoots were..…instead of adventitious shot were
Line 143. Adventitious shoots instead adventitious shoot
Data analysis section
Linea 178-179:
Rooting rate was calculated as the number of new absorbing roots divided by number of sprouting root segnments *100. ¿Were the rooting rates showed in figure 2, calculated with the same formula?
Results Section
I have several questions
- What is the meanning of ” Rooting rate of root absorbed or Rooting rate of absorbing root”? Would you mean “ absorbing roots induction/formation rate” or percentage of
- Was the rooting rate of absorbing root and rooting rate of hormone-treated adventitious shoots estimated in the same way as that of absorbing roots??
Section 3.2
I have the same question as above (Rooting rate of absorbing roots). It sounds wear to me. If I understood well, absorbing roots are developed along the root segment. Therefore is not the rooting rate of absorbing roots. In my opinion is the rate of absorbing root formation. Please redefine this term.
You should clarify these points.
Line 226: Conversely, at a constant segment length, larger diameter segmets were associated with lower rooting rates, would you mean: at a constant segment length, rates of absorbing roots was inversely correlated to the diameter segment. Please check the data provided in the text (73,33-87,18%) related to rates of absorbing roots of segments of 0.5 cm, as they do not corresponded to the data presented in table 4.
Line 240: Legend of table 4: In my opinion the term seedlings is applied to plants germinated from seeds. These is not he case, I would use the term of plantlets instead of seedlings. Moreover the epigraphe “rooting rate of root absorbed” should be modified. And also along the text
Line 262: Legend of table 5: ..and tickness of covering soil on” root propagation induction rate”
In my opinion it should be…”on adventitious shoots induction rate…”
Section 3.3
Line 260: Please check the data of shoot induction rate: 86,7 % is shown in table 5 for 3m soil tockness, but in the text it says 67%
Section 3.4
Line 274. Absorption??
Table 7: Similar as before. Please change “rooting rate of absorbing roots %”
Please check the data shown in table 7 and in the text
Section 3.7
Line 322: Shots instead of shoot
Discussion Section
The authors have recently published a paper describing an effcientin vitro protocol for shoot induction from root explants ( Ling et al., 2021; An efficient in vitro propagation protocol for direct organogenesis from root explants of a multi-purpose plant, Broussonetia papyrifera (L.) L’H´er. ex Ven). Althogh authors mentioned this work on the introductiion I think it should be included in the discussion to compare the feasibility of using root explants as starting material for propagation of this especies.
Conclusions
Please explain why segments 1.1-1.5 diameter (20-30 cm length) perfom better tan those of 1.5-2 cm
Although english is not my mother tong, I think the language / spelling and style edition in English should be reviewed.
Reviewer 2 Report
The manuscript entitled " An efficient propagation system through root cuttings of an ecological and economic value plant Broussonetia papyrifera (L.) L’Hér. ex Vent" is important for propagation of tree using root cuttings. However, the explanations and presentations needs to be revised as follows:
1) Introduction: I guess use of the stem cuttings for tree propagation is more popular than the root cuttings. Please explain why the stem cuttings are not useful in this tree.
2) Materials and Methods: The authors use the terms; proliferation coefficient, net proliferation coefficient, total proliferation coeficient, net multiplication coefficient, total multiplication coefficient. Please explain the definitions and equations of these terms in Materials and Methods.
Table 1: The title should be changed to
"Treatment of substrate and root segments with different disinfecting combinations", because the table contains substrate. Sterilization is not appropriate, because some bacteria or fungi might remain on and in the root segments.
Results: The numbers are mostly used by 4 digit such as 53.33%, but the digit number should be less than 3 (53.3%) or 2(53%).
Figure 1: The explanations should be added in this figure.
The meanings of D (0.5,0.8,,1.0,1.5) and D (5,10,20,30), Induction rate, net multiplication coefficient, total multiplication coefficient.
Line 184 "Text" is "Test".
Table 8: Please add the treatments of Tests.
Figure 2: The most effective concentration of NAA is 0.2, but the effect of IBA seems to be increasing by the addition of higher concentration than 0.5.
Figure 4: Please add the arrow head in the part of photos explained in the text.
Round 2
Reviewer 1 Report
Dear Authors,
All my suggestions have been nicely adressed.
Best regards
Reviewer 2 Report
The manuscript has been well revised, but Figure 1 needs more explanations.
Add the explanation of (c) in the legend of figure 1.
The unit of the vertical axis is missing. Please add it.
In the text, the authors refer to Figure 1 (c), (b), and (a) in this sequence. Please change the text or Figure 1.
Please add the explanation L/D means interaction between length and diameter etc.
